# The Effects of Green Tea (*Camellia sinensis*), Bamboo Extract (*Bambusa vulgaris*) and Lactic Acid on Sebum Production in Young Women with Acne Vulgaris Using Sonophoresis Treatment

**DOI:** 10.3390/healthcare10040684

**Published:** 2022-04-05

**Authors:** Karolina Chilicka, Aleksandra M. Rogowska, Monika Rusztowicz, Renata Szyguła, Antoniya Yanakieva, Binnaz Asanova, Sławomir Wilczyński

**Affiliations:** 1Department of Health Sciences, Institute of Health Sciences, University of Opole, 45-040 Opole, Poland; monika.rusztowicz@uni.opole.pl (M.R.); renata.szygula@uni.opole.pl (R.S.); 2Department of Social Sciences, Institute of Psychology, University of Opole, 45-052 Opole, Poland; arogowska@uni.opole.pl; 3Department of HTA, Faculty of Public Health, Medical University of Sofia, 1427 Sofia, Bulgaria; antoniya.yanakieva@gmail.com; 4Medical College Yordanka Filaretova, Medical University of Sofia, 1606 Sofia, Bulgaria; b.asanova@mc.mu-sofia.bg; 5Department of Basic Biomedical Science, Faculty of Pharaceutical Sciences in Sosnowiec, Medical University of Silesia in Katowice, 41-200 Sosnowiec, Poland; swilczynski@sum.edu.pl

**Keywords:** green tea extract, bamboo extract, lactic acid, acne vulgaris, sonophoresis

## Abstract

People struggling with acne vulgaris, not only experience skin eruptions and skin pain, but also report that their quality of life is worse compared with healthy people. This study examined, for the first time, the effect of sonophoresis on select skin parameters (sebum level) in young women suffering from acne vulgaris. The study included 60 women 19–23 years of age (*M* = 21.45, *SD* = 0.91) with mild and moderate facial acne. The inclusion criteria were 19–23 years of age, female or male gender, mild to moderate acne, no dermatological treatment within last 12 months, and no hormonal contraception (women). No men volunteered for the study, so the group was homogeneous. The patients were divided into two groups. Group A underwent a sonophoresis procedure using ultrasound and ultrasound gel combined with a green tea, bamboo extract ampule, and 5% lactic acid. Group B was the placebo group, where sonophoresis was performed using only ultrasound gel (no ampules). The members of the placebo group were told that they were undergoing sonophoresis with a green tea, bamboo extract, and 5% lactic acid ampule. Before and after the series of procedures, sebum levels were measured in the skin. Each patient underwent a series of five procedures using sonophoresis equipment at one-week intervals. Sonophoresis with green tea, bamboo extract, and 5% lactic acid contributed to the reduction of skin eruptions and sebum levels in the participants of the study (group A). The study results demonstrated that the combined use of plant preparations, lactic acid, and ultrasound had a positive effect on the skin of people suffering from acne vulgaris, including reduction of skin eruptions and sebum levels on the surface of the skin.

## 1. Introduction

Acne vulgaris is the most common disease of the sebaceous glands and is associated with the overproduction of sebum. It may affect up to 90% of people aged 11–25 [1]. Factors that may influence the development of the disease include genetic factors, excessive activity of the sebaceous glands, hormonal factors, the *Cutibaterium acnes* bacteria (formerly known as *Propionibacterium acnes*), excessive keratosis of sebaceous glands, superinfection with bacteria of other types *(Staphylococcus aureus*), presence of lipophilic yeasts (*Malassezia furfur*), and excessive activity of androgens [2]. Lesions that may occur due to acne include whiteheads, blackheads, pustules, nodules, cysts, and papules, which are most often located in the T zone (forehead, nose, beard), the pre-bridge area, and the seborrheic gutter between the shoulder blades. Additionally, symptoms such as pain, irritation, and skin redness may occur [3,4]. Acne can be treated dermatologically in two ways: externally with ointments or creams (retinoids, benzoyl peroxide, antibiotics, salicylic acid); and generally, with antibiotics or isotretinoin, a derivative of vitamin A. Isotretinoin has keratolytic and antibacterial properties, reduces seborrhea, and inhibits leukocyte chemotaxis [5]. After completion of acne therapy, scars and discoloration often remain, which can have a negative impact on the well-being of patients [6].

Green tea is obtained from both the leaves and buds of the *Camellia sinensis* plant. The active ingredients of green tea are polyphenols, which have antimicrobial, anti-inflammatory, and antineoplastic properties [7]. Owing to their tannin and caffeine content, they also reduce swelling and constrict blood vessels. Polyphenols are also antioxidants and play a role in preventing the oxidative damage caused by ROS [8]. Saric et al. searched the PubMed database and could only identify eight articles related to the effect of green tea and other polyphenols on sebum production and reduction of acne lesions [7]. Research conducted, among others, by Mielnik, Monfrecola, and Agamia showed that ‘there is increased phosphoinositide 3-kinase-Akt-mammalian target of rapamycin complex 1 (PI3K-Akt-mTORC1) signaling in the skin of patients with acne vulgaris and that increased PI3K-Akt-mTORC1 signaling induces sebum production.’ Plant sterols derived from *Camellia sinensis* have demonstrated a therapeutic role in the treatment of acne vulgaris [9,10,11,12]. In 2012, Im et al. showed that treatment of insulin growth factor (IGF-1)-stimulated SZ95 sebocytes with the polyphenol Epigallocatechin gallate (EGCG) led to decreased mTOR phosphorylation and, thus, mTORC1 activity [13]. Bamboo is a plant native to Asia, and its young shoots contain silica, amino acids, proteins, tyrosine, arginine, and flavonoids. Bamboo extract has antibacterial properties and protects the skin against fungi. It also has a soothing effect; therefore, it can often be found in cosmetics that are used externally, for example, in the form of creams [14].

AHAs are organic acids with one hydroxyl group attached to the alpha position of the acid. AHAs, including glycolic acid, lactic acid, malic acid, tartaric acid, and citric acid, are often used in cosmetic formulations. Moreover, lactic acid can be used at between 1 and 10% in cosmetics, and is an excellent moisturizer [15]. At between 30 and 50%, it is used in strong peels, because of its exfoliating properties. Lactic acid has an antibacterial function and is also a pH regulator. This acid effectively unblocks the skin pores and is used in anti-acne therapy (designed to prevent skin inflammations) [16]. This acid is also known in the treatment of melasma. Sharquire et al. conducted a study on 24 patients with melasma using lactic acid. The study showed that this chemical peel is effective and safe, and it was also effective as Jassner’s solution [17].

Ultrasounds are elastic waves, inaudible to the human ear, with a frequency of 16 kHz to 10 GHz [18]. Application of an active acoustic wave in physical matter causes biological, physical, and chemical changes within it. In cosmetology, ultrasounds with a frequency of 750 kHz–3 MHz and an intensity of 0.5–2.0 W/cm^2^ are applied. They mainly cause micro massage of tissue elements, leading to local hyperemia and a beneficial effect on the functioning of the lymphatic system. Previous studies that looked at the effect of ultrasound on the plasma membranes of erythrocytes showed an increase in their permeability (change in the resting potential of cell membranes). In addition, ultrasounds contribute to an increase in the kinetic energy of the preparation molecules used during the treatment, as well as to improvement of the skin’s blood supply, which results in better absorption of active ingredients. In sonophoresis, the active preparation is combined with the ultrasound gel and applied to the treatment section before performing the procedure. Drug penetration is most likely in the 1 to 2 mm depth range. Sonophoresis increases the absorption of substances and activates cellular metabolism. This method is proven to be much more effective than the external application of cosmetics [19,20,21].

We did not find any studies related to the use of sonophoresis in acne vulgaris in the PubMed database. However, this procedure was used, among others, in cosmetology to eliminate other skin defects. Zasada et al. described the effect of using sonophoresis, as well as 0.3 and 0.5% of retinol (in liquid crystal formula) on mature skin [22]. In 2016, Park et al. used sonophoresis in combination with fractional radiofrequency to introduce 5-aminolevulinic acid (ALA) into the skin of three domestic swine. The scientists concluded that the synergy of both methods largely contributed to the excellent results that were produced. In 2019, the same researchers used ultrasound with glutamic acid, demonstrating that this type of treatment is promising in the cosmetic and therapeutic fields [23,24]. 

There are many cosmetic treatments that reduce the secretion of sebum, including cosmetic acids. Chilicka et al., in their research published in 2020 in the *Scientific Reports* journal, showed that the use of both azalaic and pyruvic acid peels led to sebum reduction on the skin surface [25]. Fąk et al. observed an immediate reduction in sebum levels after microdermabrasion [26]. Colchicine has also been proposed for the management of acne [27]. Laser and light devices have been proposed in the management of acne-related scars, with variable results [28]. Another method applied to reduce acne lesions is the Nd:YAG laser. Monib et al. compared the effect of the use of IPL equipment and a Nd:YAG laser. The improvement in total lesions was significant in the Nd:YAG group but nonsignificant in the IPL Group (*p* < 0.001, *p* = 0.13, respectively). These procedures may cause temporary side effects, in the form of irritation, redness, and itching that may last 2–3 days [29]. These side effects do not occur with sonophoresis; therefore, it could be a promising alternative for people who do not want to hideaway from the world for a few days because of any after-effects. It is also a relatively short procedure, which takes up to 10 min.

## 2. Materials and Methods

### 2.1. Study Design

A single blind placebo study with follow-up analysis was performed to test the effect of sonophoresis on selected skin parameters in young, adult women. The exclusion criteria for treatment (both group A and B) were as follows: pregnancy, lactation, skin active inflammatory, allergic, fungal, bacterial skin diseases, recent surgical procedures in the treatment areas, active herpes, allergy to any components of cosmetics, hormonal contraception, external and internal isotretinoin use, active rosacea, eczema, psoriasis, numerous telangiectasias, pacemaker, heart problems, implants (metal, silicone, or saline), active tuberculosis, severe acne and propensity to keloids, or any dermatological treatment (antibiotics). The inclusion criteria for this examination (both group A and B) were 19–23 years of age, female or male gender, mild to moderate acne, no dermatological treatment within the last 12 months, and no use of hormonal contraception (women). Figure 1 is a consort flow chart of the clinical study group. Ninety-seven people submitted to the study, 37 of whom were rejected due to the exclusion criteria. Women 19–23 years of age were included in the procedure. The group was homogeneous because no men entered the study. Sixty women suffering from acne vulgaris were divided into two groups. They were assigned to the groups on the basis of a random selection of envelopes containing a certain group designation (A or B). Group A was the group that received sonophoresis using the ampule and ultrasound gel, while group B was the placebo group where only the ultrasound gel without the ampule was used. Each group consisted of an equal number of participants (A: *n* = 30, B: *n* = 30). Individuals from group B were not informed that it was the placebo group and were under the impression that they were undergoing a full sonophoresis procedure.

The study was conducted at the Institute of Health Sciences of University of Opole, Poland from September to December 2020, and Medical College Yordanka Filaretova, Medical University of Sofia, Sofia, Bulgaria at February to March 2020 and June to July 2021.

Any other cosmetic procedures, as well as the use of new cosmetics or sebum-regulating cosmetics, were forbidden during the study period. Only cosmetics such as micellar water and moisturizing creams were allowed. The respondents were also asked to discontinue any supplementation that might alter the sebum levels in the skin (yeast tablets, sulfur tablets, or herbal teas). The research was approved by the Human Research Ethics Committee of the Opole Medical School (No. KB/59/NOZ/2019), according to the principles of the Declaration of Helsinki. The study was registered at https://www.isrctn.com (No. ISRCTN98261720)- accessed on 3 January 2020. Patients signed voluntary written consent before starting the study (including the photography material), and they were informed that they could withdraw from the study at any time, without giving a reason.

### 2.2. Participants

Sixty young women, 19–23 years of age (*M* = 21.45, *SD* = 0.91), participated in the study. Moderate acne severity was diagnosed in 58 (97% of the total sample) individuals, while two people (3%) showed mild acne. The mean duration of acne was 6.20 years (with a range between 2 and 12 years). The majority of the group used antibiotics and a minority used isotretinoin. Most of the women were of normal weight (85%). In the last week, the majority of participants reported drinking alcohol (58%), eating sweets (95%), and not smoking (65%). More details on the demographic characteristics of the study sample are presented in Table 1.

### 2.3. Measures

#### 2.3.1. Acne Vulgaris

The severity of acne vulgaris was assessed using the global acne grading system (GAGS), which was devised in 1997 by Doshi, Zaheer, and Stiller. This scale divides the whole body into areas, which include the forehead, cheeks, nose, chin, chest, and back, and assigns a factor to each area on the basis of size [30,31]. The location and factors are forehead—2; right cheek—2; left cheek—2; nose—1; chin—1; chest and upper back—3. Each type of lesion is given a value depending on severity: no lesions = 0, comedones = 1, papules = 2, pustules = 3, and nodules = 4. The score for each area (local score) is calculated using the formula Local score = Factor × Grade (0–4). Acne severity is graded using the global score, which is the sum of local scores. A score of 1–18 is considered mild; 19–30, moderate; 31–38, severe; and >39, very severe.

#### 2.3.2. Skin Parameters

Skin measurements were taken three times: before the treatment series, week 1 and weeks 2 after the end of the fifth procedure. The participants were asked to remove their make-up in the evening the day before the measurements and not to apply any cosmetics to their faces. The test was carried out in the morning hours (between 8 and 12 am), and the conditions in the room were as follows: 20–24 degrees Celsius and humidity of about 40–50%. Before the skin measurements were started, the participants were acclimated to the room for 25 min. Skin sebum measurements were assessed in four places on the face: between the eyebrows, 1 cm from the right and left nostril, and 1 cm under the lower lip (chin). A Derma Unit SCC 3 apparatus (Courge & Khazaka, Cologne, Germany) was used for the test, which allows measuring pH (skin-pH-meter), hydration (Corneometer), and greasing via sebum level (Sebumeter). Skin sebum level is based on photometry. The measuring head resembles a cassette that contains a 0.1 mm thick synthetic matte foil pressed against the measured area with a uniform force of 10 N. Before applying the cassette to the skin, it should be reset on the device and then applied to the tested area. It should be kept still for 30 s, signaled by the computer software clock. After 30 s, the cassette is reinserted into the machine. The photocell determines the degree of foil transparency in the Sebumeter, and the result is the difference in foil transparency before and after skin contact. Measurement results are in the sebum range of 0–350 μg/cm^2^. If the device displays a result of 0, the skin is devoid of sebum [32].

#### 2.3.3. Demographic Survey

The patients were asked nine questions:How old are you?What is your height (in cm)?How much do you weigh (in kg)?How many years have you been suffering from acne?Have you ever been treated with antibiotics meant to reduce acne (for example, tetracycline)?Have you ever been treated with isotretinoin?Have you smoked cigarettes in the last week?Have you drinking alcohol in the past week?Have you eaten sweets in the last week?

Questions 5–9 were answered with a YES or NO.

### 2.4. Treatment Procedure (Intervention)

The severity of acne lesions was measured using the global acne grading system (GAGs). Both groups, A (experimental) and B (placebo), had sonophoresis every seven days (total five treatments), but group A had the actual procedure performed using an ultrasound-generating apparatus, the ultrasound gel (composition: polyacrylate substrate, preservatives, purified water) from Żelpol, Poland, and ampules (INCI composition: Aqua, Green tea extract, *Bambusa vulgaris* (bamboo) extract, Lactic Acid, Mimose Extract, Phenoxyethanol, Caprylyl Glycol, Sal Maris) from Apis Cosmetics, Poland. Group B also underwent a procedure using the ultrasound generator and ultrasound gel, but without the cosmetic preparation. However, this group was unaware that they received the placebo treatment. The treatment protocol for both groups was identical. The difference was that in group B, an ampule was not added to the ultrasound gel. Before the sonophoresis procedure was performed, make-up in the treatment area was removed using micellar fluid by Apis Cosmetics, Poland. Its composition is (INCI) Aqua, Polyglyceryl-4 Caprate, Glycerin, Sodium Hyaluronate, Mimose Extract, Aloe Extract, Phenoxyethanol, and Caprylyl Glycol. Facial make-up removal was performed by applying micellar fluid to a cotton pad and washing the entire treatment area. Then the ultrasound gel combined with the ampule was applied to the face in group A. For this, the ampule and the ultrasound gel were poured into a cup and applied with a brush. The preparation was introduced into the skin (5 mL ampule and 10 mL ultrasound gel) using a sonophoresis apparatus by Hebe Poland and a special spatula. In group B, only the ultrasound gel (10 mL) was applied, without the ampule. The gel was introduced into the skin by ultrasound, just as in group A. The parameters were the same in all 5 procedures for groups A and B, with ultrasound power at 0.25 W/cm^2^ for a duration of 10 min. After the procedure, the face was washed to remove any remnants of the preparation using the same micellar fluid that had been used to remove the make-up before the procedure. For home care, micellar water and moisturizing cream were recommended (no matting or sebo-regulating preparations were allowed, so as not to disturb the test results). Cosmetic treatments were not permitted to be performed on the facial area during the study period and 2 weeks after its completion, which is when the second post-treatment measurement was performed. Using a solarium or swimming pool was also not permissible, nor taking any anti-acne medications or dietary supplements that could affect the secretion of sebum. 

### 2.5. Statistical Analysis

In the first step, descriptive analyses were performed, such as the analysis of the distribution of variables due to measures of central tendencies (mean, standard deviation, median, modal, skewness, and kurtosis) and the Kolmogorov-Smirnov *d* test for normality of the distribution. The descriptive statistics indicated good statistical properties, and, therefore, in a subsequent step, parametric one-way ANOVA with repeated measures was used to examine the differences between the experimental and control groups with respect to acne severity and facial skin greasing at baseline, as well as seven and fourteen days after sonophoresis treatment. The Tuckey honest significant difference (HSD) test was used for post-hoc analysis between the groups. The prevalence of mild and moderate acne in the experimental and control groups, before and after sonophoresis treatment, was also compared using Pearson’s chi-squared (χ^2^) test of independence. All statistical tests were performed using STATISTICA 13.1 software.

## 3. Results

### 3.1. Changes in Acne Severity after Sonophoresis Treatment

A one-way repeated measures ANOVA was conducted to compare the effect of sonophoresis treatment on acne severity. Acne severity was measured twice using GAGS: (1) before sonophoresis treatment; and (2) 2 weeks after completion of all five treatments. Differences in acne severity between the first and second measurements were assessed in the experimental and control groups. Table 2 shows descriptive statistics, such as range, mean (*M*), and standard deviation (*SD*). For acne severity assessed before and after sonophoresis treatment using GAGS, there was a significant group effect (Experimental vs. Control) on acne (*F*(1, 58) = 36.14, *p* < 0.001, η_p_^2^ = 0.38), and treatment conditions (Before and After sonophoresis) on acne severity (*F*(1, 58) = 110.24, *p* < 0.001, η_p_^2^ = 0.65). An interaction effect was also found between groups and treatment conditions (*F*(1, 58) = 110.24, *p* < 0.001, η_p_^2^ = 0.65). Although the control group did not differ from the experimental group before treatment, the posthoc test showed a significant reduction in acne after sonophoresis treatment in the experimental group (*p* < 0.001).

As seen in Figure 2, moderate acne was diagnosed in the entire experimental group (*n* = 30, 100%), and in the majority of women in the control group (*n* = 28, 93%). The difference in acne severity between the groups was not statistically significant before sonophoresis treatment, χ^2^(1) = 2.07, *p* = 0.15, ϕ = 0.18. However, after sonophoresis treatment, a significant difference in the frequency of mild and moderate acne (χ^2^(1) = 35.62, *p* < 0.001, ϕ = 0.77) was found between the experimental and control groups. Mild acne was diagnosed in most participants in the experimental group (*n* = 25, 83%), and in only two participants in the control group (7%). Figure 3 shows the acne severity in participants in the two groups, before sonophoresis treatment and two weeks after finishing treatment.

### 3.2. Changes in Facial Skin Greasing after Sonophoresis

The effect of sonophoresis treatment on facial skin greasing was examined using one-way repeated measures ANOVA. Facial skin greasing was measured three times (i.e., before treatment, and seven and fourteen days after finishing the treatment) in three areas of the face: (1) between the eyebrows, (2) around the nose, and (3) around the bottom lip. 

Differences in acne severity between the first (at baseline), second (one week after finishing the treatment), and third (two weeks after finishing the treatment) measurements were compared between the experimental (treated) and control (placebo) groups. Means (*M*), standard deviations (*SD*), and one-way ANOVA with repeated measures for facial skin greasing are shown in Table 3 and Figure 4. There was a significant medium effect of the group parameter (experimental vs. control) on facial greasing around the nose (*F*(1, 58) = 6.15, *p* < 0.05, η_p_^2^ = 0.10). The placebo group showed higher levels of greasing than the experimental group (*p* < 0.05). However, the effect of the treatment group on skin greasing between the eyebrows and around the bottom lip was not statistically significant. A large ‘time’ effect was found, when the measurement of facial greasing (between the eyebrows, around the nose, and bottom lip) was performed before, and seven and fourteen days after sonophoresis treatment (see Table 3 and Figure 4 for more details). The first skin assessment (at baseline) differed significantly (*p* < 0.001) from the second (week 1 after treatment finishing) and third ones (weeks 2 after treatment). Furthermore, an interaction effect was found between the groups and time (Table 3). Facial skin greasing was higher at baseline, while it was lower after treatment (week 1 and weeks 2 after finishing), in the experimental group when compared to the control sample (*p* < 0.01).

## 4. Discussion

To our knowledge, sonophoresis treatment with green tea, bamboo extract, and 5% lactic acid, with respect to cosmetological treatment of acne vulgaris, was evaluated for the first time in this study. The results of this study demonstrate that sonophoresis is a safe procedure, without side effects such as skin irritation or redness, and does not necessitate a recovery period. Both sebum levels and the amount of skin eruptions were significantly reduced after the series of procedures. 

Sonophoresis is a procedure that allows a preparation to penetrate deeper into the skin, which results in improving the flow of blood and lymph, and also the regeneration of cells [22]. While there are studies in the literature on external use, as well as dietary supplementation, of green tea, there are no articles on the effects of sonophoresis combined with green tea. Mahmood et al. conducted a study on 22 non-smoking men aged 22–28 years and found that the use of a topical green tea and lotus extract decreased facial sebum production [33]. In 2016, Lu et al. conducted a study to see whether oral supplementation with green tea could improve skin conditions and reduce skin eruptions in people with acne. The study included 80 people and found a significant decrease in inflammatory lesions among people who received 1500 mg of decaffeinated green tea for four weeks, compared with people who received a placebo of cellulose capsules [34]. In 2012, Jung et al. conducted a study in Korea, in which a topical green tea extract (polyphenol-60) was found to decrease open comedones and the number of pustules, but did not improve closed comedones in people with acne vulgaris [35]. This study also shows that topical green tea extract can be helpful, but the sonophoresis had a positive effect on the reduction of all skin eruptions. In their study, Sharquie et al. observed a significant improvement in the reduction of skin eruptions in acne vulgaris patients given a 2% tea lotion, compared with patients given a 5% zinc sulfate solution (both treatments involved the use of the products twice a day for two months), who showed no statistically significant changes. The type of tea used in the study was not specified [36]. Elsaie et al. conducted a study on 20 women and men aged 15–36 who suffered from acne, and found that participants had reduced acne lesions after using a green tea lotion twice a day for 6 weeks [37]. In that study, they observed 20 patients; therefore, a lower sample size and no control group, compared to our study, but the reduction of acne lesions after using a green tea lotion was observed. In 2010, Mahmood et al. tested the use of 3% GT over an eight-week period in a group of ten healthy men, observing that sebum production was significantly reduced (*p* < 0.05) after treatment. Specifically, sebum production had decreased by nearly 10% in the first week and as much as 60% by week 8. The study limitation was the small sample size and no placebo group [38]. Our study was conducted on women, and the sample size was large, and also there was a placebo group. Facial skin greasing was higher at baseline, while lower after treatment (week 1 and week 2 after finishing) in the experimental group, when compared to the control sample (*p* < 0.01).

With respect to the use of sonophoresis in cosmetology, there are articles on the effects of ultrasound on skin aging. A study by Hsin-Ti et al. showed that women who received the antioxidant Lycogen during sonophoresis had significantly fewer signs of skin aging than women who did not receive Lycogen [39]. In another study, Zasada et al. used sonophoresis to inject 0.3 and 0.5% retinol deep into the skin. The procedure resulted in a significant reduction (from baseline) in mean moisture, level of sebum, hyperpigmentation, and erythema [22]. Choi et al. performed a study to analyze and compare the effectiveness of ablative fractional laser (FXL) pretreatment and/or sonophoresis for enhancing the penetration of 5-aminolevulinic acid. The study did not show any additional positive effects of the laser treatment on ALA penetration [40]. In a study by Jung et al., a combination of ultrasound and heat was shown to increase the percutaneous penetration of L-ascorbic acid [41]. In a study by Lodhi et al., bamboo extract accelerated cutaneous wound healing in rats [42]. In addition, a 2016 paper by Gong et al. described the anti-inflammatory effect of bamboo [14]. Moreover, Davane et al. claimed that the flavonoids in bamboo have been documented to have antibacterial activity [43].

The results from this study show that sonophoresis with the use of green tea, bamboo extract, and 5% lactic acid might positively reduce skin eruption and decrease skin sebum in people suffering from acne vulgaris. It should be added that sonophoresis is a cosmetological treatment and cannot displace dermatological methods. It can support other dermatological therapies (drugs and topical therapies) to reduce sebum levels and skin eruptions in people with mild to moderate acne vulgaris [44,45]. This study also showed that facial skin greasing was lowered after treatment (week 1 and week 2 after completion). The treatment also reduced the number of skin eruptions with the bamboo extract, owing to its anti-inflammatory effect.

Green tea polyphenols have proven activity versus a broad spectrum of microbes. Green tea polyphenolic catechins: EGCG and ECG, have been shown to inhibit the growth of a broad range of Gram-negative and Gram-positive bacterial species with moderate potency. No adverse effects have been associated with green tea (oral consumption) [46]. Mathew et al. showed that the protective and repairing effect of green tea (jejunal mucosa of rats) after fasting-induced failure is dependent on cell proliferation and the induction of specific GF (growth factors) [47]. Lodhi et al., in their study, showed that ethyl acetate fraction of *B. vulgaris* leaves inhibits paw edema and accelerates cutaneous wound healing (anti-inflammatory and antioxidant potential) [42].

Using green tea, bamboo extract, and lactic acid is the right solution for patients with AV. Green tea has a broad spectrum of microbes, bamboo extract has anti-inflammatory properties, and lactic acid has antibacterial and proliferation functions, and is also a pH regulator.

### Study Limitations

The study used a Derma Unit SSC3 device to measure select skin parameters, such as sebum. In the future, additional measurements such as pH and moisture of the skin could be included, as well as the use of a specialized camera to take photos to improve the testing aspects of the study. For example, the Visiopor^®^ PP 34 camera uses a specific UV light to visualize fluorescing acne lesions with an 8 × 6.4 mm minimum area. The other limitation of the study was the research cohort, as the study included a homogeneous group of young women. In the future, a more extensive study sample should be targeted that consists of both genders and a more comprehensive age range. Furthermore, it would be valuable to compare the effectiveness of sonophoresis in combination with other cosmetic herbs. In the future, a longer measurement period could also be used; for example week 1, week 2, week 3, and week 4, to check the difference between them. In the future, we would like to conduct further research, where in the control group we would use active treatment without sonophoresis. Thus, we could compare the articles; this could show if sonophoresis is needed for achieving the treatment effect seen.

## 5. Conclusions

This study was the first to evaluate the effects of using green tea, bamboo extract, and 5% lactic acid with sonophoresis. The results show that this approach is a safe and effective treatment for acne vulgaris. It demonstrated that the treatment had a positive effect on reducing skin eruptions and sebum on the skin’s epidermis. However, it should be emphasized that a cosmetic procedure can in no way replace dermatological therapies that specifically treat severe forms of acne. It can only be used as an additional treatment for people with mild to moderate acne vulgaris or excessive sebum secretion.

## Figures and Tables

**Figure 1 healthcare-10-00684-f001:**
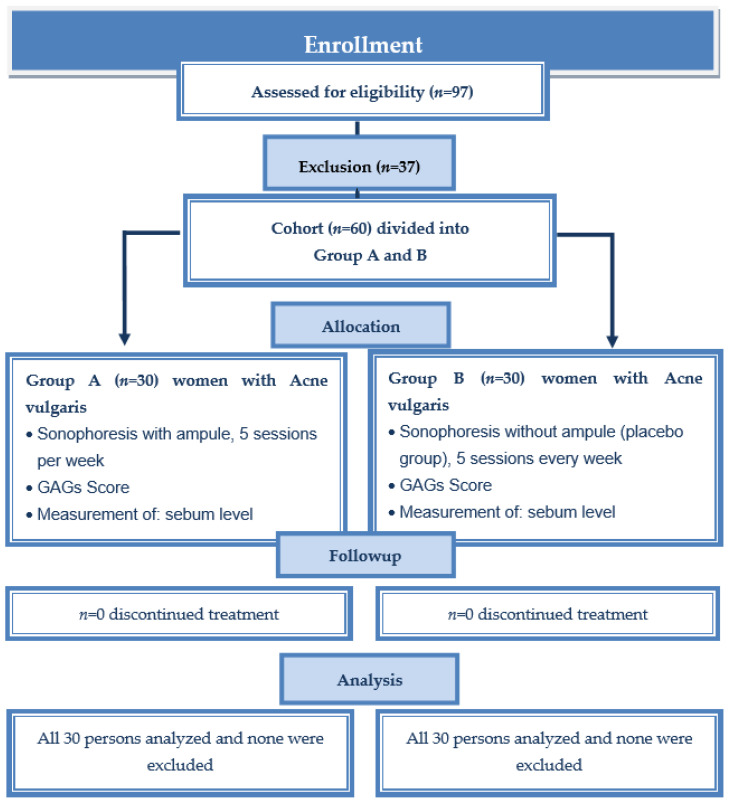
Consort flow chart of clinical study group A and group B (placebo).

**Figure 2 healthcare-10-00684-f002:**
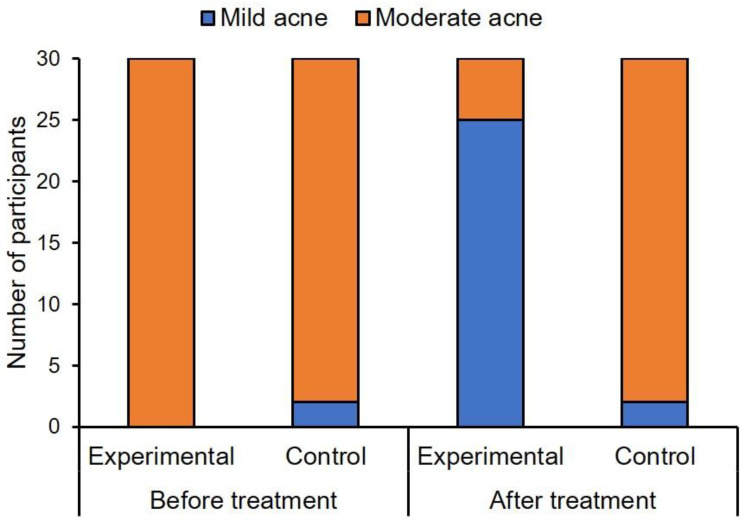
Differences in acne severity between the experimental and control groups, before and after sonophoresis treatment.

**Figure 3 healthcare-10-00684-f003:**
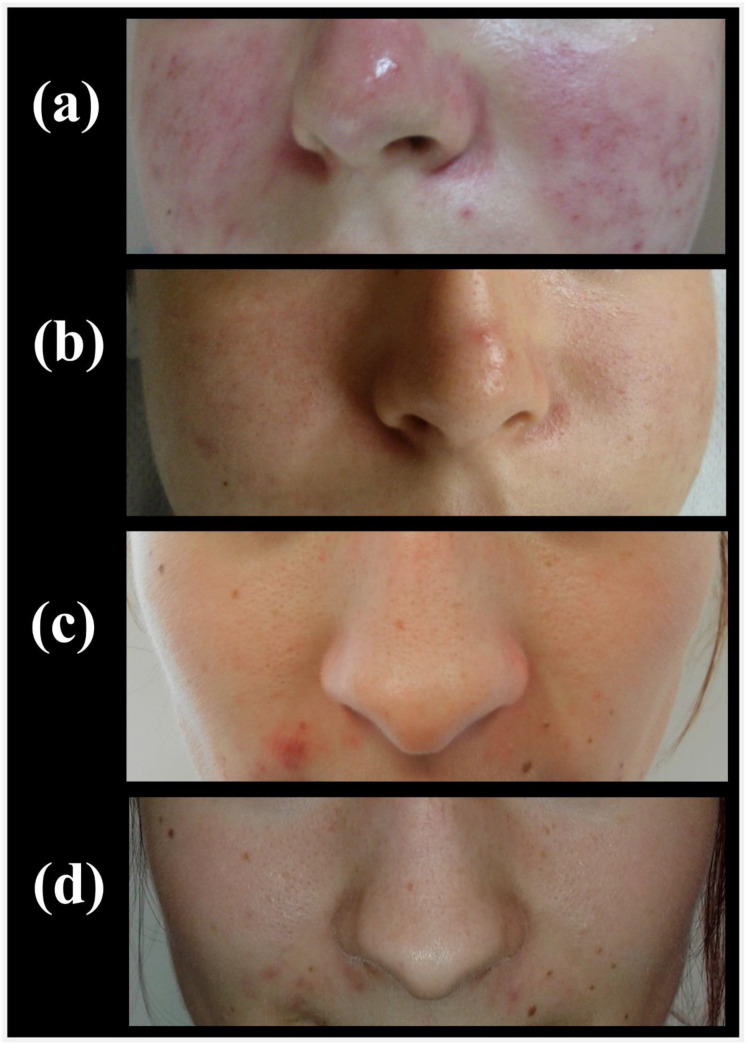
Acne severity in a participant from the experimental group: (**a**) before treatment, and (**b**) 2 weeks after treatment; and in a participant from the control group: (**c**) before treatment, and (**d**) 2 weeks after treatment.

**Figure 4 healthcare-10-00684-f004:**
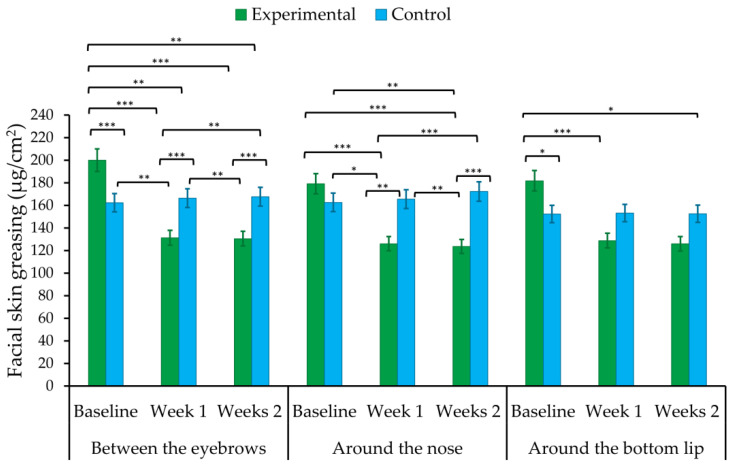
Mean scores of skin greasing at baseline, week 1, and weeks 2 after completion of sonophoresis treatment in the experimental and control groups. * *p* < 0.05, ** *p* < 0.01, *** *p* < 0.001.

**Table 1 healthcare-10-00684-t001:** Demographic characteristics of the sample.

Variable		Total Sample		Experimental Group		Control Group				
	Range	*n*	%	*n*	%	*n*	%	χ^2^	*p*	φ
Age (years) (*M, SD*)	19–23	21.45	0.91	21.17	0.99	21.73	0.74	330.50 *_U_*	0.059	−0.27 *_d_*
Height (cm) (*M, SD*)	153–187	164.77	5.44	163.83	5.51	165.70	5.29	373.00 *_U_*	0.257	−0.1 *_d_*
Weight (kg) (*M, SD*)	44–87	61.22	8.40	61.83	7.61	60.60	9.21	520.50 *_U_*	0.300	0.16 *_d_*
Body mass index (BMI) (*M, SD*)	17.40–31.96	22.54	2.81	23.01	2.38	22.06	3.15	584.50 *_U_*	0.048	0.30 *_d_*
Normal	18.51–24.99	51	85.00	26	86.67	25	83.33	1.52	0.468	0.16 *_V_*
Overweight	25.00–29.99	8	13.33	3	10.00	5	16.67			
Obese	30.00<	1	1.67	1	3.33	0	0.00			
Acne duration (years) (*M, SD*)	2–12	6.20	2.15	6.57	1.14	5.83	2.81	558.00 *_U_*	0.106	0.24 *_d_*
Acne severity (GAG) (*M, SD*)	17–25	21.10	1.68	21.50	1.64	20.70	1.66	562.50 *_U_*	0.092	0.25 *_d_*
Mild	1–18	2	3.33	0	0.00	2	6.67	2.07	0.150	−0.19
Moderate	19–30	58	96.67	30	100.00	28	93.33			
Antibiotics use (ever)										
Yes		43	71.67	26	86.67	17	56.67	6.65	0.010	−0.33
No		17	28.33	4	13.33	13	43.33			
Isotretinoin use (ever)										
Yes		16	26.67	11	36.67	5	16.67	3.07	0.080	−0.23
No		44	73.33	19	63.33	25	83.33			
Smoking cigarettes last week										
Yes		21	35.00	10	33.33	11	36.67	0.07	0.787	0.04
No		39	65.00	20	66.67	19	63.33			
Drinking alcohol last week										
Yes		35	58.33	18	60.00	7	23.33	8.297	0.004	0.37
No		25	41.67	12	40.00	23	76.67			
Eating sweets last week										
Yes		57	95.00	30	100.00	27	90.00	3.16	0.076	−0.23
No		3	5.00	0	0.00	3	10.00			

Note. *_U_* = Mann–Whitey *U*-test for independent samples, *_d_* = Cohen’s *d* effect size, *_V_* = Cramer’s *V* for effect size.

**Table 2 healthcare-10-00684-t002:** Descriptive statistics for acne severity in the GAGS.

	Experimental Group			Control Group		
Condition	Range	*M*	*SD*	Range	*M*	*SD*
Before treatment	19–25	21.50	1.63	17–24	20.70	1.66
After treatment	9–19	14.73	3.04	17–24	20.70	1.62

**Table 3 healthcare-10-00684-t003:** Means, standard deviations, and one-way ANOVA with repeated measures for facial skin greasing.

	Experimental		Control					
Facial Skin Greasing	*M*	*SD*	*M*	*SD*	Effect	*F ratio*	*df*	η_p_^2^
Between the eyebrows								
Baseline	199.98	30.82	162.43	53.79	G	2.20	1.58	0.04
Week 1 after treatment	131.35	25.22	166.40	34.01	T	49.08 ***	2.116	0.46
Weeks 2 after treatment	130.52	23.31	167.70	31.35	G × T	64.20 ***	2.116	0.53
Around the nose								
Baseline	179.23	44.53	162.67	51.98	G	6.15 *	1.58	0.10
Week 1 after treatment	126.03	27.01	165.67	45.57	T	24.92 ***	2.116	0.30
Weeks 2 after treatment	123.69	25.37	172.30	45.65	G × T	40.15 ***	2.116	0.41
Around the bottom lip								
Baseline	181.83	40.33	152.33	50.76	G	0.65	1.58	0.01
Week 1 after treatment	128.83	27.33	153.30	48.64	T	29.64 ***	2.116	0.34
Weeks 2 after treatment	126.00	23.26	152.53	34.48	G × T	30.87 ***	2.116	0.35

Note. *n* = 30. ANOVA = analysis of variance; Experimental = sonophoresis treatment group; Control = placebo group; G = group; T = time; G × T = interaction between group and time. * *p* < 0.05, *** *p* < 0.001.

## Data Availability

Not applicable.

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
