# Peer review of "The Effects of Green Tea (Camellia sinensis), Bamboo Extract (Bambusa vulgaris) and Lactic Acid on Sebum Production in Young Women with Acne Vulgaris Using Sonophoresis Treatment"

_healthcare, 2022, doi:10.3390/healthcare10040684_

Round 1

Reviewer 1 Report

interesting article but really not clear the comparison of the group since you would like to show that sonophoresis is interesting to improve efficacy of a mix ingredient . in this case we would have better to confirm a group with the mixt compare to a group with the mixt + sonophoresis ... to keep blind you will have to use sonophoresis without open it just doing the gesture shame. 

however the article is interesting since in acne real solution are rare . 

the corrections are now fine for publication 

Author Response

Dear Editor and Reviewer

We would like to thank the Editor and Reviewer for their time and valuable comments. Your suggested edits and constructive criticism were invaluable, and we hope that this version of the manuscript will be better. We sincerely appreciate your suggestions and did our best to incorporate them into the revised manuscript. Our responses are highlighted in blue.

English language and style

(x) English language and style are fine/minor spell check required

  • A native English-speaking editor has revised the English language and style.

Comments and Suggestions for Authors

  1. Interesting article but really not clear the comparison of the group since you would like to show that sonophoresis is interesting to improve the efficacy of a mixed ingredient. In this case, we would have better to confirm a group with the mixed compare to a group with the mixt + sonophoresis ... to keep blind you will have to use sonophoresis without open it just doing the gesture shame. 
  • Thank You very much for your offer. Certainly, in the future study, we will use this kind of placebo group.

Reviewer 2 Report

The article,” The Effects of Green Tea (Camellia Sinensis) and Bamboo Extract (Bambusa Vulgaris) on Sebum Production in Young Women with Acne Vulgaris using Sonophoresis Treatment” by Chilicka et al, has investigated the effect of sonophoresis with a green tea and bamboo extract ampule on acne vulgaris. This is well designed study. Some editing for grammar is needed to improve the clarity. 

Author Response

Dear Editor and Reviewer

We would like to thank the Editor and Reviewer for their time and valuable comments. Your suggested edits and constructive criticism were invaluable, and we hope that this version of the manuscript will be better. We sincerely appreciate your suggestions and did our best to incorporate them into the revised manuscript. Our responses are highlighted in blue.

English language and style

(x) Extensiveediting of English language and style required

A native English speaking editor has revised the English language and style.

  1. The article,” The Effects of Green Tea (Camellia Sinensis) and Bamboo Extract (Bambusa Vulgaris) on Sebum Production in Young Women with Acne Vulgaris using Sonophoresis Treatment” by Chilicka et al, has investigated the effect of sonophoresis with a green tea and bamboo extract ampule on acne vulgaris. This is well-designed study. Some editing for grammar is needed to improve the clarity. 

  • Thank You for the comments. We made the editing for grammer and improved the clarity.

Reviewer 3 Report

A very interesting original study showing that sonophoresis  in  the green tea , bamboo extract and  lactic acid group contributed to the reduction of skin eruptions and sebum levels in the participants of the study. The paper already underwent multiplòe rounds of revisions, so only minor queries are made before considering the paper acceptable:

line 55 you could add: "Colchicine has also been proposed in the management of acne" and cite: doi: 10.3390/pharmaceutics14020294.

Line 57 you could add: "laser and lights devices have been proposed in the management of acne-related scars, with variable results" and cite: doi:10.1007/s10103-020-03063-6

Good Luck!

Author Response

Dear Editor and Reviewer

We would like to thank the Editor and Reviewer for their time and valuable comments. Your suggested edits and constructive criticism were invaluable, and we hope that this version of the manuscript will be better. We sincerely appreciate your suggestions and did our best to incorporate them into the revised manuscript. Our responses are highlighted in blue.

English language and style

(x) English language and style are fine/minor spell check required

A native English speaking editor has revised the English language and style.

A very interesting original study showing that sonophoresis  in  the green tea , bamboo extract and  lactic acid group contributed to the reduction of skin eruptions and sebum levels in the participants of the study. The paper already underwent multiplòe rounds of revisions, so only minor queries are made before considering the paper acceptable:

  1. line 55 you could add: "Colchicine has also been proposed in the management of acne" and cite: doi: 10.3390/pharmaceutics14020294.
  • Thank You very much for this comment. We addend this sentence in the manuscript.
  1. Line 57 you could add: "laser and lights devices have been proposed in the management of acne-related scars, with variable results" and cite: doi:10.1007/s10103-020-03063-6
  • Thank You very much for this comment. We addend this sentence in the manuscript.

This manuscript is a resubmission of an earlier submission. The following is a list of the peer review reports and author responses from that submission.

Round 1

Reviewer 1 Report

The article,” The Effects of Green Tea (Camellia Sinensis) and Bamboo Extract (Bambusa Vulgaris) on Sebum Production in Young Women with Acne Vulgaris using Sonophoresis Treatment” by Chilicka et al, has investigated the effect of sonophoresis with a green tea and bamboo extract ampule on acne vulgaris. This study may provide insight into novel therapeutic targets of acne vulgaris. This study design is straight forward. It is a well written and interesting article. I have few comments:

The ampules used in sonophoresis composed of Green Tea, Bamboo Extract and Lactic Acid. Lactic acid has been used for treatment of pimples, acne, photoaged skin or hyper pigmentation. More citation should be added on beneficial effect of lactic acid in introduction section.

The time after the treatment or sonophoresis is presented as weeks or days (7 days/1 week, 14 days/2 weeks) in different part of manuscript, which is quite confusing.  It should be kept in one format.

No significant difference was detected in facial greasing (eyebrows, around the nose, and bottom lip) between 7days (week1) or 14 days (weeks2) after treatment. This should be discussed.

Authors have detected the effect of sonophoresis with green tea and bamboo extract on skin eruptions and sebum content in the epidermis of subjects with acne. Also, other group of subjects receiving sonophoresis with ultrasound gel only were included as control. Is there any previous study on effect of topical application of combination of green tea and bamboo extract lotions on acnes and sebum production?  If not, it will be interesting if we could compare the effect of combination of green tea and bamboo extract by topical application alone and with sonophoresis on  acne.

In study design part of method section, the word, “peace maker”, should be corrected.

Cutibacterium acne  is a major bacterium linked to the skin condition of acne.  The bamboo extract has antibacterial properties. Is there any previous investigation showing effect of bamboo extract on growth of C. acnes?

What could be exact role of bamboo extract and lactic acid in the mixture with green tea in treating acne. It should be discussion and relevant citation should be provided.

 Citation should be added on effect of Green Tea and Bamboo Extract on proliferation and toxicity of keratinocytes.  

Reviewer 2 Report

Interesting study but the english should be improve// do not use "we" when speaking about your proposition but indirect style. 

the introduction is confused 

The results are not clearly present , no stat present comparing control/ experimental at baseline (should be >0.05 no different) and at end of the study (should be <0.05 different at least) to confirm effacy. 

Reviewer 3 Report

Minor suggestion for improvement:

Acne vulgaris is the most common skin disease that usually begins during puberty and can continue into later adulthood. The study presented in the manuscript was done accurately and demonstrated the results of the use of sonophoresis with Green Tea and Bamboo Extract for treatment of mild acne vulgaris: the sonophoresis treatment using plant extracts does not cause any irritation and is a completely non-invasive procedure, comparing it to e.g. lasers; it does not cause side effects and a long recovery time; in placebo group (treated with sonophoresis without plant extract) showed no improvement over the control group. The use of sonophoresis and plant extracts resulted in a reduction in the number of skin eruptions, a reduction in skin inflammation, and a reduction in the amount of sebum secreted. This is an additional advantage because the excess sebum on the surface of the skin causes these people to add powder, which additionally clogs the sebaceous glands. This contributes to the deterioration of the skin condition. Cosmetological treatments are not a substitute for pharmacological treatment, but are an option to alleviate mild acne vulgaris.

I recommend the authors of this manuscript in their next study to compare the effectiveness of this method with that of conventional synthetic drugs for topical treatment of mild acne.

In paragraph 55 - replace s in staphylococcus by capital letter. 

Round 2

Reviewer 1 Report

The article,” The Effects of Green Tea (Camellia Sinensis) and Bamboo Extract (Bambusa Vulgaris) on Sebum Production in Young Women with Acne Vulgaris using Sonophoresis Treatment” by Chilicka et al, has investigated the effect of sonophoresis with a green tea and bamboo extract ampule on acne vulgaris. This study provides insight into novel therapeutic targets of acne vulgaris. This study design is straight forward. It is a well witten article.

Significance should be added in bar graph in figure 4.

Is the word “peacemaker”, misspelled for "pacemaker"?

Author Response

Dear Editor and Reviewer

We would like to thank the Editor and REviewer for their time and valuable comments. Our responses are highlighted in blue.

  1. English language and style: We changed it in all article.
  2. Significance should be added in bar graph in figure 4: we changed it
  3. Is the word "peacemaker" misspelled for "pacemaker: we changed it.
